# BayesDiff: Estimating Pixel-wise Uncertainty in Diffusion via Bayesian Inference

**Siqi Kou[1], Lei Gan[4], Dequan Wang[1,5], Chongxuan Li[2,3]\*, and Zhijie Deng[1]\***

[1]Qing Yuan Research Institute, SEIEE, Shanghai Jiao Tong University
[2]Gaoling School of Artificial Intelligence, Renmin University of China
[3]Beijing Key Laboratory of Big Data Management and Analysis Methods
[4]School of Computer Science, Fudan University [5]Shanghai Artificial Intelligence Laboratory
`happy-karry@sjtu.edu.cn`, `21307130211@m.fudan.edu.cn`
`dequanwang@sjtu.edu.cn`, `chongxuanli1991@gmail.com`, `zhijied@sjtu.edu.cn`

## Abstract

Diffusion models have impressive image generation capability, but low-quality generations still exist, and their identification remains challenging due to the lack of a proper sample-wise metric. To address this, we propose *BayesDiff*, a pixel-wise uncertainty estimator for generations from diffusion models based on Bayesian inference. In particular, we derive a novel uncertainty iteration principle to characterize the uncertainty dynamics in diffusion, and leverage the last-layer Laplace approximation for efficient Bayesian inference. The estimated pixel-wise uncertainty can not only be aggregated into a sample-wise metric to filter out low-fidelity images but also aids in augmenting successful generations and rectifying artifacts in failed generations in text-to-image tasks. Extensive experiments demonstrate the efficacy of BayesDiff and its promise for practical applications. Our code is available at `https://github.com/karrykkk/BayesDiff`.

## 1 Introduction

The ability of diffusion models to gradually denoise noise vectors into natural images has paved the way for numerous applications, including image synthesis (Dhariwal & Nichol, 2021; Rombach et al., 2022), image inpainting (Lugmayr et al., 2022), text-to-image generation (Saharia et al., 2022; Gu et al., 2022; Zhang et al., 2023), etc. However, there are still inevitable low-quality generations causing poor user experience in downstream applications. A viable remediation is to filter out low-quality generations, which, yet, cannot be trivially realized due to the lack of a proper metric for image quality identification. For example, the traditional metrics such as the Fréchet Inception Distance (FID) (Heusel et al., 2017) and Inception Score (IS) (Salimans et al., 2016) scores estimate the distributional properties of the generations instead of sample-wise quality.

Bayesian uncertainty has long been used to identify data far from the manifold of training samples (Maddox et al., 2019; Deng et al., 2021). The notion is intuitive—the Bayesian posterior delivers low uncertainty for the data witnessed during training while high uncertainty for the others. This fits the requirement that the generations returned to the users should be as realistic as the training images. However, the integration of Bayesian uncertainty and diffusion models is not straightforward. Diffusion models typically involve large-scale networks, necessitating efficient Bayesian inference strategies. Additionally, the generation of images typically involves an intricate reverse diffusion process, which adds to the challenge of accurately quantifying their uncertainty.

To address these challenges, we propose BayesDiff, a framework for estimating the pixel-wise Bayesian uncertainty of the images generated by diffusion models. We develop a novel uncertainty iteration principle that applies to various sampling methods to characterize the dynamics of pixel-wise uncertainty in the reverse diffusion process, as illustrated in Figure 1. We leverage the last-layer Laplace approximation (LLLA) (Kristiadi et al., 2020; Daxberger et al., 2021a) for efficient Bayesian inference of pre-trained score models. Finally, BayesDiff enables the simultaneous

---
\*Corresponding authors.

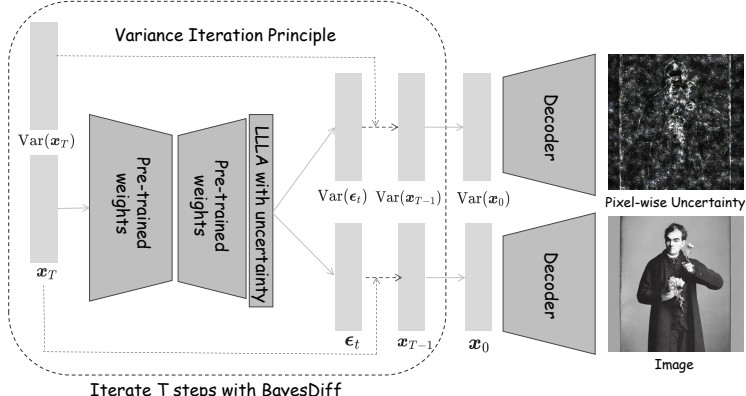

Figure 1: Given an initial point $\boldsymbol{x}_T \sim \mathcal{N}(\mathbf{0}, \mathbf{I})$, our BayesDiff framework incorporates uncertainty into the denoising process and generates images with pixel-wise uncertainty estimates.

delivery of image samples and pixel-wise uncertainty estimates. The naive BayesDiff can be slow due to the involved Monte Carlo estimation, so we further develop an accelerated variant of it to enhance efficiency.

We can aggregate the obtained pixel-wise uncertainty into image-wise metrics (e.g., through summation) for generation filtering. Through extensive experiments conducted on ADM (Dhariwal & Nichol, 2021), U-ViT (Bao et al., 2023), and Stable Diffusion (Rombach et al., 2022) using samplers including DDIM (Song et al., 2020) and DPM-Solver (Lu et al., 2022), we demonstrate the efficacy of BayesDiff in filtering out images with cluttered backgrounds. Moreover, in text-to-image generation tasks, we surprisingly find that pixel-wise uncertainty can enhance generation diversity by augmenting good generations, and rectify failed generations containing artifacts and mismatching with textual descriptions. Additionally, we perform comprehensive ablation studies to seek a thorough and intuitive understanding of the estimated pixel-wise uncertainty.

## 2 BACKGROUND

This section briefly reviews the methodology of diffusion probabilistic models and introduces Laplace approximation (LA), a classic method for approximate Bayesian inference.

### 2.1 DIFFUSION MODELS

Let $\boldsymbol{x} \in \mathbb{R}^{c \times h \times w}$ denote an image. A diffusion model (DM) (Ho et al., 2020) typically assumes a forward process gradually diffusing data distribution $q(\boldsymbol{x})$ towards $q_t(\boldsymbol{x}_t), \forall t \in [0, T]$, with $q_T(\boldsymbol{x}_T) = \mathcal{N}(\mathbf{0}, \tilde{\sigma}^2 \mathbf{I})$ as a trivial Gaussian distribution. The transition distribution obeys a Gaussian formulation, i.e., $q_t(\boldsymbol{x}_t|\boldsymbol{x}) = \mathcal{N}(\boldsymbol{x}_t; \alpha_t \boldsymbol{x}, \sigma_t^2 \mathbf{I})$, where $\alpha_t, \sigma_t \in \mathbb{R}^+$. The reverse process is defined with the data score $\nabla_{\boldsymbol{x}_t} \log q_t(\boldsymbol{x}_t)$, which is usually approximated by $-\epsilon_\theta(\boldsymbol{x}_t, t)/\sigma_t$ with $\epsilon_\theta$ as a parameterized noise prediction network trained by minimizing:

$$\mathbb{E}_{\boldsymbol{x} \sim q(\boldsymbol{x}), \boldsymbol{\epsilon} \sim \mathcal{N}(\mathbf{0}, \mathbf{I}), t \sim \mathcal{U}(0, T)}[w(t)\|\epsilon_\theta(\alpha_t \boldsymbol{x} + \sigma_t \boldsymbol{\epsilon}, t) - \boldsymbol{\epsilon}\|_2^2] \tag{1}$$

where $w(t)$ denotes a weighting function.

Kingma et al. (2021) show that the stochastic differential equation (SDE) satisfying the transition distribution specified above takes the form of

$$d\boldsymbol{x}_t = f(t)\boldsymbol{x}_t dt + g(t)d\boldsymbol{\omega}_t, \tag{2}$$

where $\boldsymbol{\omega}_t$ is the standard Wiener process, $f(t) = \frac{d \log \alpha_t}{dt}$, and $g(t)^2 = \frac{d\sigma_t^2}{dt} - 2\frac{d \log \alpha_t}{dt}\sigma_t^2$. An SDE and an ordinary differential equation (ODE) starting from $\boldsymbol{x}_T$ capturing the reverse process can be constructed as (Song et al., 2021):

$$d\boldsymbol{x}_t = [f(t)\boldsymbol{x}_t + \frac{g(t)^2}{\sigma_t}\boldsymbol{\epsilon}_t]dt + g(t)d\bar{\boldsymbol{\omega}}_t \tag{3}$$

$$\frac{d\boldsymbol{x}_t}{dt} = f(t)\boldsymbol{x}_t + \frac{g(t)^2}{2\sigma_t}\boldsymbol{\epsilon}_t, \tag{4}$$

where $\bar{\boldsymbol{\omega}}_t$ is the reverse-time Wiener process when time flows backward from $T$ to $0$, $dt$ is an infinitesimal negative timestep and $\boldsymbol{\epsilon}_t := \epsilon_\theta(\boldsymbol{x}_t, t)$ denotes the noise estimated by the neural network (NN) model. We could sample $\boldsymbol{x}_0 \sim q(\boldsymbol{x})$ from $\boldsymbol{x}_T \sim q_T(\boldsymbol{x}_T)$ by running backwards in time with the numerical solvers (Lu et al., 2022; Karras et al., 2022) for Equation (3) or Equation (4).

## 2.2 Bayesian Inference in Deep Models and Laplace Approximation

Bayesian inference turns a deterministic neural network into a Bayesian neural network (BNN). Let $p(\mathcal{D}|\theta)$ denote the data likelihood of the NN $f_\theta$ for the dataset $\mathcal{D} = \{(\boldsymbol{x}^{(n)}, y^{(n)})\}_{n=1}^N$. Assuming an isotropic Gaussian prior $p(\theta)$, BNN methods estimate the Bayesian posterior $p(\theta|\mathcal{D}) = p(\theta)p(\mathcal{D}|\theta)/p(\mathcal{D})$, where $p(\mathcal{D}|\theta) := \prod_n p(y^{(n)}|\boldsymbol{x}^{(n)}, \theta) = \prod_n p(y^{(n)}|f_\theta(\boldsymbol{x}^{(n)}))$, and predict for new data $\boldsymbol{x}^*$ with $p(y|\boldsymbol{x}^*, \mathcal{D}) = \mathbb{E}_{p(\theta|\mathcal{D})}p(y|f_\theta(\boldsymbol{x}^*))$.

Due to NNs' high nonlinearity, analytically computing $p(\theta|\mathcal{D})$ is often infeasible. Hence, approximate inference techniques such as variational inference (VI) (Blundell et al., 2015; Hernández-Lobato & Adams, 2015; Louizos & Welling, 2016; Zhang et al., 2018; Khan et al., 2018), Laplace approximation (LA) (Mackay, 1992; Ritter et al., 2018), Markov chain Monte Carlo (MCMC) (Welling & Teh, 2011; Chen et al., 2014; Zhang et al., 2019), and particle-optimization based variational inference (POVI) (Liu & Wang, 2016) are routinely introduced to yield an approximation $q(\theta) \approx p(\theta|\mathcal{D})$. Among them, LA has recently gained particular attention because it can apply to pre-trained models effortlessly in a post-processing manner and enjoy strong uncertainty quantification (UQ) performance (Foong et al., 2019; Daxberger et al., 2021a;b).

LA approximates $p(\theta|\mathcal{D})$ with $q(\theta) = \mathcal{N}(\theta; \theta_{\text{MAP}}, \boldsymbol{\Sigma})$, where $\theta_{\text{MAP}}$ denotes the maximum a posteriori (MAP) solution $\theta_{\text{MAP}} = \arg\max_\theta \log p(\mathcal{D}|\theta) + \log p(\theta)$, and $\boldsymbol{\Sigma} = [-\nabla^2_{\theta\theta}(\log p(\mathcal{D}|\theta) + \log p(\theta))|_{\theta=\theta_{\text{MAP}}}]^{-1}$ characterizes the Bayesian uncertainty over model parameters. Last-layer Laplace approximation (LLLA) (Kristiadi et al., 2020; Daxberger et al., 2021a) further improves the efficiency of LA by concerning only the parameters of the last layer of the NN. It is particularly suited to the problem of uncertainty quantification for DMs, as DMs are usually large.

## 3 Methodology

We incorporate LLLA into the noise prediction model in DMs for uncertainty quantification at a single timestep. We then develop a novel algorithm to estimate the dynamics of pixel-wise uncertainty along the reverse diffusion process. We also develop a variant of it for practical acceleration.

### 3.1 Laplace Approximation on Noise Prediction Model

Usually, the noise prediction model is trained to minimize Equation (1) under a weight decay regularizer, which corresponds to the Gaussian prior on the NN parameters. Namely, we can regard the pre-trained parameters as a MAP estimation and perform LLLA. Of note, the noise prediction problem corresponds to a regression under Gaussian likelihood, based on which we estimate the Hessian matrix involved in the approximate posterior (or its variants such as the generalized Gauss-Newton matrix). The last layer in DMs is often linear w.r.t. the parameters, so the Gaussian approximate posterior distribution on the parameters directly leads to a Gaussian posterior predictive:

$$p(\boldsymbol{\epsilon}_t|\boldsymbol{x}_t, t, \mathcal{D}) \approx \mathcal{N}(\epsilon_\theta(\boldsymbol{x}_t, t), \text{diag}(\gamma^2_\theta(\boldsymbol{x}_t, t))), \tag{5}$$

where we abuse $\theta$ to denote the parameters of the pre-trained DM. We keep only the diagonal elements in the Gaussian covariance, $\gamma^2_\theta(\boldsymbol{x}_t, t)$, because they refer to the pixel-wise variance of the predicted noise, i.e., the pixel-wise prediction uncertainty of $\boldsymbol{\epsilon}_t$. Implementation details regarding the LLLA are shown in Appendix A.4.

### 3.2 Pixel-wise Uncertainty Estimation in Reverse Denoising Process

Next, we elaborate on integrating the uncertainty obtained above into the reverse diffusion process.

Although various sampling methods may correspond to various reverse diffusion processes, the paradigm for characterizing the uncertainty dynamics is similar. Take the SDE-form one (Equation (3)) for example, introducing Bayesian uncertainty to the noise prediction model yields:

$$d\boldsymbol{x}_t = [f(t)\boldsymbol{x}_t + \frac{g(t)^2}{\sigma_t}\boldsymbol{\epsilon}_t]dt + g(t)d\bar{\boldsymbol{\omega}}_t, \tag{6}$$

where $\boldsymbol{\epsilon}_t \sim \mathcal{N}(\epsilon_\theta(\boldsymbol{x}_t, t), \mathrm{diag}(\gamma_\theta^2(\boldsymbol{x}_t, t)))$. Assume the following discretization for it:

$$\boldsymbol{x}_{t-1} = \boldsymbol{x}_t - (f(t)\boldsymbol{x}_t + \frac{g(t)^2}{\sigma_t}\boldsymbol{\epsilon}_t) + g(t)\boldsymbol{z}, \tag{7}$$

where $\boldsymbol{z} \sim \mathcal{N}(\mathbf{0}, \mathbf{I})$. To estimate the pixel-wise uncertainty of $\boldsymbol{x}_{t-1}$, we apply variance estimation to both sides of the equation, giving rise to

$$\mathrm{Var}(\boldsymbol{x}_{t-1}) = (1 - f(t))^2\mathrm{Var}(\boldsymbol{x}_t) - (1 - f(t))\frac{g(t)^2}{\sigma_t}\mathrm{Cov}(\boldsymbol{x}_t, \boldsymbol{\epsilon}_t) + \frac{g(t)^4}{\sigma_t^2}\mathrm{Var}(\boldsymbol{\epsilon}_t) + g(t)^2\mathbf{1}, \tag{8}$$

where $\mathrm{Cov}(\boldsymbol{x}_t, \boldsymbol{\epsilon}_t) \in \mathbb{R}^{c \times w \times h}$ denotes the pixel-wise covariance between $\boldsymbol{x}_t$ and $\boldsymbol{\epsilon}_t$. With this, we can iterate over it to estimate the pixel-wise uncertainty of the final $\boldsymbol{x}_0$. Recalling that $\mathrm{Var}(\boldsymbol{\epsilon}_t) = \gamma_\theta^2(\boldsymbol{x}_t, t)$, the main challenges then boils down to estimating $\mathrm{Cov}(\boldsymbol{x}_t, \boldsymbol{\epsilon}_t)$.

**The estimation of $\mathrm{Cov}(\boldsymbol{x}_t, \boldsymbol{\epsilon}_t)$.** By the law of total expectation $\mathbb{E}(\mathbb{E}(X|Y)) = \mathbb{E}(X)$, there is

$$\begin{aligned}
\mathrm{Cov}(\boldsymbol{x}_t, \boldsymbol{\epsilon}_t) &= \mathbb{E}(\boldsymbol{x}_t \odot \boldsymbol{\epsilon}_t) - \mathbb{E}\boldsymbol{x}_t \odot \mathbb{E}\boldsymbol{\epsilon}_t \\
&= \mathbb{E}_{\boldsymbol{x}_t}(\mathbb{E}_{\boldsymbol{\epsilon}_t|\boldsymbol{x}_t}(\boldsymbol{x}_t \odot \boldsymbol{\epsilon}_t|\boldsymbol{x}_t)) - \mathbb{E}\boldsymbol{x}_t \odot \mathbb{E}_{\boldsymbol{x}_t}(\mathbb{E}_{\boldsymbol{\epsilon}_t|\boldsymbol{x}_t}(\boldsymbol{\epsilon}_t|\boldsymbol{x}_t)) \\
&= \mathbb{E}_{\boldsymbol{x}_t}(\boldsymbol{x}_t \odot \epsilon_\theta(\boldsymbol{x}_t, t)) - \mathbb{E}\boldsymbol{x}_t \odot \mathbb{E}_{\boldsymbol{x}_t}(\epsilon_\theta(\boldsymbol{x}_t, t))
\end{aligned} \tag{9}$$

where $\odot$ denotes the element-wise multiplication. To estimate this, we need the distribution of $\boldsymbol{x}_t$.

We notice that it is straightforward to estimate $\mathbb{E}(\boldsymbol{x}_t)$ via a similar iteration rule to Equation (8):

$$\mathbb{E}(\boldsymbol{x}_{t-1}) = (1 - f(t))\mathbb{E}(\boldsymbol{x}_t) - \frac{g(t)^2}{\sigma_t}\mathbb{E}(\boldsymbol{\epsilon}_t). \tag{10}$$

Given these, we can reasonably assume $\boldsymbol{x}_t$ follows $\mathcal{N}(\mathbb{E}(\boldsymbol{x}_t), \mathrm{Var}(\boldsymbol{x}_t))$, and then $\mathrm{Cov}(\boldsymbol{x}_t, \boldsymbol{\epsilon}_t)$ can be approximated with Monte Carlo (MC) estimation:

$$\mathrm{Cov}(\boldsymbol{x}_t, \boldsymbol{\epsilon}_t) \approx \frac{1}{S}\sum_{i=1}^{S}(\boldsymbol{x}_{t,i} \odot \epsilon_\theta(\boldsymbol{x}_{t,i}, t)) - \mathbb{E}\boldsymbol{x}_t \odot \frac{1}{S}\sum_{i=1}^{S}\epsilon_\theta(\boldsymbol{x}_{t,i}, t), \tag{11}$$

where $\boldsymbol{x}_{t,i} \sim \mathcal{N}(\mathbb{E}(\boldsymbol{x}_t), \mathrm{Var}(\boldsymbol{x}_t)), i = 1, \ldots, S$.

**Applying our method to existing samplers.** The derivation from Equation (7) to Equation (11) presents our general methodology based on the classical Euler sampler of reverse-time SDE, which can be applied to an arbitrary existing sampler of diffusion models in principle. For broader interests and simplicity, we show the explicit rules for DDPM (Ho et al., 2020) as an instance.

Specifically, the sampling rule of DDPM (Ho et al., 2020) is:

$$\mathbf{x}_{t-1} = \frac{1}{\sqrt{\alpha_t'}}(\mathbf{x}_t - \frac{1 - \alpha_t'}{\sqrt{1 - \bar{\alpha}_t'}}\boldsymbol{\epsilon}_t) + \sqrt{\beta_t}\mathbf{z} \tag{12}$$

where $\alpha_t' := 1 - \beta_t$ with $\beta_t$ as the given noise schedule for DDPM and $\bar{\alpha}_t' = \prod_{s=1}^{t}\alpha_s'$. Using the same techniques as above, we can trivially derive the corresponding iteration rules:

$$\mathrm{Var}(\mathbf{x}_{t-1}) = \frac{1}{\alpha_t'}\mathrm{Var}(\mathbf{x}_t) - 2\frac{1 - \alpha_t'}{\alpha_t'\sqrt{1 - \bar{\alpha}_t'}}\mathrm{Cov}(\boldsymbol{x}_t, \boldsymbol{\epsilon}_t) + \frac{(1 - \alpha_t')^2}{\alpha_t'(1 - \bar{\alpha}_t')}\mathrm{Var}(\boldsymbol{\epsilon}_t) + \beta_t \tag{13}$$

$$\mathbb{E}(\boldsymbol{x}_{t-1}) = \frac{1}{\sqrt{\alpha_t'}}\mathbb{E}(\boldsymbol{x}_t) - \frac{1 - \alpha_t'}{\sqrt{\alpha_t'(1 - \bar{\alpha}_t')}}\mathbb{E}(\boldsymbol{\epsilon}_t) \tag{14}$$

Equation (11) can still be leveraged to approximate $\mathrm{Cov}(\boldsymbol{x}_t, \boldsymbol{\epsilon}_t)$. We iterate over these equations to obtain $\boldsymbol{x}_0$ as well as its uncertainty $\mathrm{Var}(\boldsymbol{x}_0)$. Since the advanced samplers, e.g., Analytic-DPM (Bao et al., 2021), DDIM (Song et al., 2020) and DPM-Solver (Lu et al., 2022), are more efficient and

---

**Algorithm 1** Pixel-wise uncertainty estimation via Bayesian inference. (BayesDiff)

---

**Input:** Starting point $\boldsymbol{x}_T$, Monte Carlo sample size $S$, Pre-trained noise prediction model $\epsilon_\theta$.
**Output:** Image generation $\boldsymbol{x}_0$ and pixel-wise uncertainty $\mathrm{Var}(\boldsymbol{x}_0)$.
 1: Construct the pixel-wise variance prediction function $\gamma_\theta^2$ via LLLA;
 2: $\mathbb{E}(\boldsymbol{x}_T) \leftarrow \boldsymbol{x}_T, \mathrm{Var}(\boldsymbol{x}_T) \leftarrow \boldsymbol{0}, \mathrm{Cov}(\boldsymbol{x}_T, \boldsymbol{\epsilon}_T) \leftarrow \boldsymbol{0}$;
 3: **for** $t = T \rightarrow 1$ **do**
 4:     Sample $\boldsymbol{\epsilon}_t \sim \mathcal{N}(\epsilon_\theta(\boldsymbol{x}_t, t), \mathrm{diag}(\gamma_\theta^2(\boldsymbol{x}_t, t)))$;
 5:     Obtain $\boldsymbol{x}_{t-1}$ via Equation (7);
 6:     Estimate $\mathbb{E}(\boldsymbol{x}_{t-1})$ and $\mathrm{Var}(\boldsymbol{x}_{t-1})$ via Equation (10) and Equation (8);
 7:     sample $\boldsymbol{x}_{t-1,i} \sim \mathcal{N}(\mathbb{E}(\boldsymbol{x}_{t-1}), \mathrm{Var}(\boldsymbol{x}_{t-1})), i = 1, \ldots, S$;
 8:     Estimate $\mathrm{Cov}(\boldsymbol{x}_{t-1}, \boldsymbol{\epsilon}_{t-1})$ via Equation (11).
 9: **end for**

---

widely adopted, we also present the explicit formulations corresponding to them in Appendix A.1. The 2-order DPM-solver is particularly popular in practice, e.g., in large-scale text-to-image models, but it is non-trivial to apply the above derivations directly to it because there is an extra hidden state introduced between $\boldsymbol{x}_t$ and $\boldsymbol{x}_{t-1}$. We propose to leverage structures like conditional independence to resolve this. Find more details in Appendix A.1.

**Continuous-time reverse process.** Instead of quantifying the uncertainty captured by the discrete diffusion process as Equation (7), we can also directly quantify that associated with the original continuous-time process, i.e., Equation (6). We have derived an approximate expression illustrating the pattern of uncertainty dynamics at arbitrary timestep in $[0, T]$. However, estimating $\mathrm{Var}(\boldsymbol{x}_0)$ using it is equally laborious as a discrete-time reverse process because discretization is still required to approximate the involved integration. See Appendix A.2 for more discussion.

**Algorithm.** Algorithm 1 demonstrates the procedure of applying the developed uncertainty iteration principle to the SDE sampler in Equation (7). After obtaining the pixel-wise uncertainty $\mathrm{Var}(\boldsymbol{x}_0)$, we can aggregate the elements into an image-wise metric for low-quality image filtering.

### 3.3 THE PRACTICAL ACCELERATION

Algorithm 1 computes the uncertainty of the hidden state at each sampling step, abbreviated as BayesDiff. The function $\gamma_\theta^2(\boldsymbol{x}_t, t)$ produces outcomes along with $\epsilon_\theta(\boldsymbol{x}_t, t)$, raising minimal added cost. Yet, the MC estimation of $\mathrm{Cov}(\boldsymbol{x}_t, \boldsymbol{\epsilon}_t)$ in Equation (11) causes $S$ (usually $S > 10$) more evaluations of $\epsilon_\theta$, which can be prohibitive in the deployment scenarios.

To address this, we develop a faster variant of BayesDiff by performing uncertainty quantification on only a subset of the denoising steps rather than all of them, dubbed BayesDiff-Skip. Concretely, we pre-define a schedule $\tilde{\boldsymbol{t}} := \{\tilde{t}_1, \ldots, \tilde{t}_U\}$ in advance. For each timestep $t$, if $t \in \tilde{\boldsymbol{t}}$, we sample $\boldsymbol{\epsilon}_t$ from LLLA and estimate corresponding uncertainty $\mathrm{Var}(\boldsymbol{x}_{t-1})$ following the uncertainty iteration principle. Otherwise, we adopt the deterministic sampling step where $\mathrm{Cov}(\boldsymbol{x}_t, \boldsymbol{\epsilon}_t)$ and $\mathrm{Var}(\boldsymbol{\epsilon}_t)$ are set to zero. We outline such a procedure in Appendix A.3.

**Consistency between BayesDiff-Skip and BayesDiff.** We check the reliability of BayesDiff-Skip here. Concretely, we generate 96 images using BayesDiff (skipping interval $= 0$) under the DDIM sampling rule on ImageNet and mark the top 9 and bottom 9 samples with the highest and lowest uncertainty according to the summation of pixel-wise uncertainty. We send the same random noises (i.e., $\boldsymbol{x}_T$) to BayesDiff-Skip (skipping interval $> 0$) with various skipping intervals (unless specified otherwise, we evenly skip) to obtain generations close to the aforementioned 96 images yet with various uncertainties. We plot the uncertainties of the marked images in Figure 2. It is shown that the marked images that BayesDiff is the most uncertain about remain the same for BayesDiff-Skip. Notably, in this experiment, BayesDiff-Skip can achieve a $5\times$ reduction in running time.

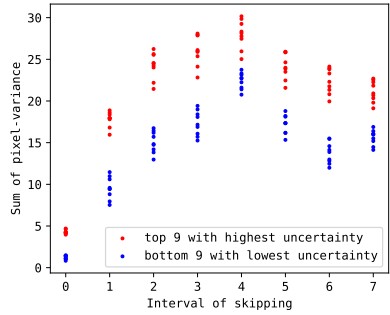

Figure 2: A study on the reliability of the BayesDiff-Skip algorithm. The top images with the highest uncertainty selected by BayesDiff are still with high uncertainty in BayesDiff-Skip algorithm.

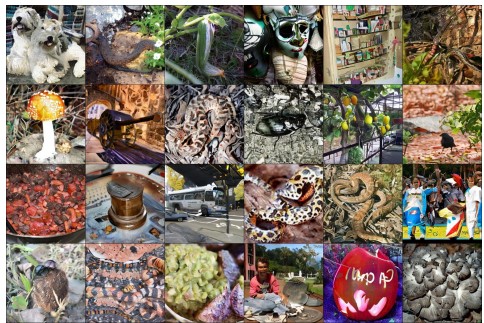 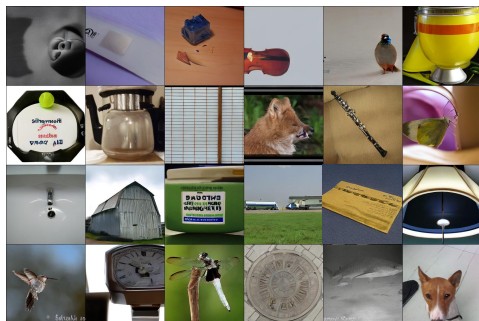

Figure 3: The images with the highest (left) and lowest (right) uncertainty among 5000 unconditional generations of U-ViT model trained on ImageNet at $256 \times 256$ resolution.

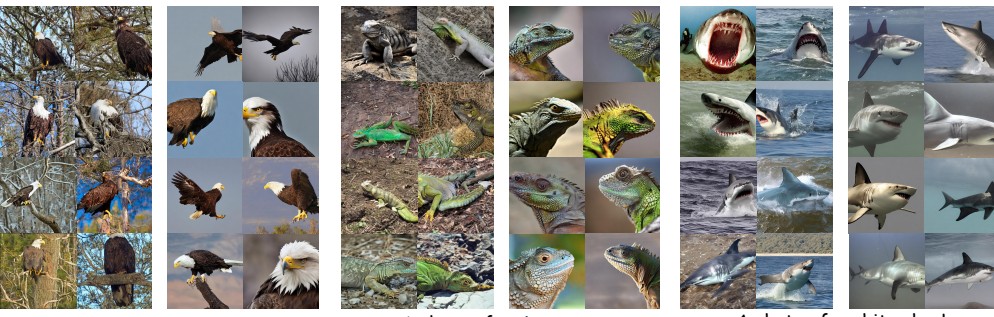

A photo of a bald eagle.   A photo of an iguana.   A photo of a white shark.

Figure 4: The images with the highest (left) and lowest (right) uncertainty among 80 generations on Stable Diffusion at $512 \times 512$ resolution.

## 4 EXPERIMENTS

In this section, we demonstrate the efficacy of BayesDiff in filtering out low-quality generations. Besides, we show that pixel-wise uncertainty can aid in generating diverse variations of the successful generation and addressing artifacts and misalignment of failure generations in text-to-image tasks. At last, we seek an intuitive understanding of the uncertainty estimates obtained by BayesDiff. We sum over the pixel-wise uncertainty to obtain an image-wise metric. Unless specified otherwise, we set the Monte Carlo sample size $S$ to 10 and adopt BayesDiff-Skip with a skipping interval of 4, which makes our sampling and uncertainty quantification procedure consume no more than $2\times$ time than the vanilla sampling method.

### 4.1 EFFECTIVENESS IN LOW-QUALITY IMAGE FILTERING

**Comparison between high and low uncertainty images.** We first conduct experiments on the U-ViT (Bao et al., 2023) model trained on ImageNet (Deng et al., 2009) and Stable Diffusion, performing sampling and uncertainty quantification using BayesDiff-Skip. The sampling algorithm follows the 2-order DPM-Solver with 50 function evaluations (NFE). We display the generations with the highest and lowest uncertainty in Figure 3 and Figure 4.

As shown above, our image-wise uncertainty metric is likely to indicate the level of clutter and the degree of subject prominence in the image. It can be used to detect low-quality images with cluttered backgrounds in downstream applications.

**Relationship between our sample-wise uncertainty metric and traditional distributional metrics.** We use BayesDiff-Skip to generate 100,000 images on CELEBA (Liu et al., 2015) based on DDPM (Ho et al., 2020) model and DDIM sampler, 250,000 $256 \times 256$ ImageNet images based on U-ViT (Bao et al., 2023) and 2-order DPM-Solver, and 250,000 $128 \times 128$ ImageNet images based on ADM (Dhariwal & Nichol, 2021) and DDIM. We separately divide the sets into five groups of the same size with descending uncertainty. We compute the traditional metrics for each group of data, including Precision (Kynkäänniemi et al., 2019), which evaluates the fidelity of the genera-

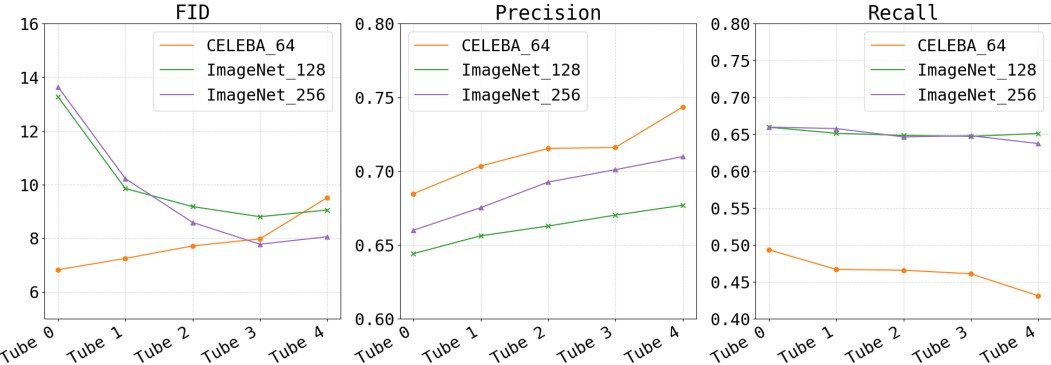

Figure 5: FID, Precision and Recall scores of 5 groups of generations with descending uncertainty on CELEBA and ImageNet datasets. Results show there is a strong correlation between our sample-wise uncertainty metric and traditional distributional metrics.

Table 1: Comparison on three metrics between randomly selected images and our selected images. We use 50 NFE for both DDIM and DPM-Solver sampler.

| Model | Dataset | Sampler | FID ↓ | | Precision ↑ | | Recall ↑ | |
|-------|---------|---------|-------|------|-------------|------|----------|------|
| | | | random | ours | random | ours | random | ours |
| ADM | ImageNet 128 | DDIM | $8.68 \pm 0.04$ | 8.48 | 0.661 | 0.665 | 0.655 | 0.653 |
| ADM | ImageNet 128 | 2-order DPM-Solver | $9.77 \pm 0.03$ | 9.67 | 0.657 | 0.659 | 0.649 | 0.649 |
| U-ViT | ImageNet 256 | 2-order DPM-Solver | $7.24 \pm 0.02$ | 6.81 | 0.698 | 0.705 | 0.658 | 0.657 |
| U-ViT | ImageNet 512 | 2-order DPM-Solver | $17.72 \pm 0.03$ | 16.87 | 0.728 | 0.732 | 0.602 | 0.604 |

tion, Recall (Kynkäänniemi et al., 2019), which accounts for the diversity, and FID, which conjoins fidelity and diversity. We present the results in Figure 5.

Notably, images with higher uncertainty have higher Recalls in both scenarios, i.e., higher diversity, and images with lower uncertainty have higher Precisions, i.e., higher fidelity. This echoes the trade-off between Precision and Recall. Moreover, as Figure 3 implies, images with high uncertainty have cluttered elements, which is consistent with the results of Precision. However, the trend of FID is different among the three datasets. The main reason is that the generations on the simple CELEBA are all good enough, so diversity becomes the main factor influencing FID. Conversely, on the more complex ImageNet, fidelity is the main factor influencing FID.

**Criterion for low-quality image filtering to improve the quality of generated distribution.** We generate 50,000 images using a 2-order DPM-Solver sampler and BayesDiff-Skip on ImageNet at $256 \times 256$ resolution to explore the distribution of image-wise uncertainty. As shown in Figure 6, the empirical distribution of the image-wise uncertainty of the generations is approximately a normal distribution. According to the 3-sigma criterion of normal distribution for eliminating outliers, we eliminate low-quality images by filtering out the images with uncertainty higher than $\mu+\sigma$, which is equivalent to top $16\%$ images with the highest uncertainty. To test the effectiveness of this criterion, we generate 60,000 images with various models and samplers and select the 50,000 images with lower uncertainty. We compute the Precision, Recall, and FID of these samples in Table 1, which also includes the random selection baseline. The results validate that we filter out low-quality images precisely with such an uncertainty-based filtering criterion.

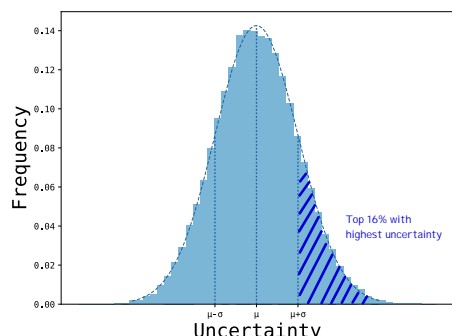

Figure 6: The empirical distribution of the uncertainty estimates yielded by our approach. The dashed line denotes the normal distribution fitted on them. Inspired by this, we propose to filter out the top $16\%$ samples with the highest uncertainty.

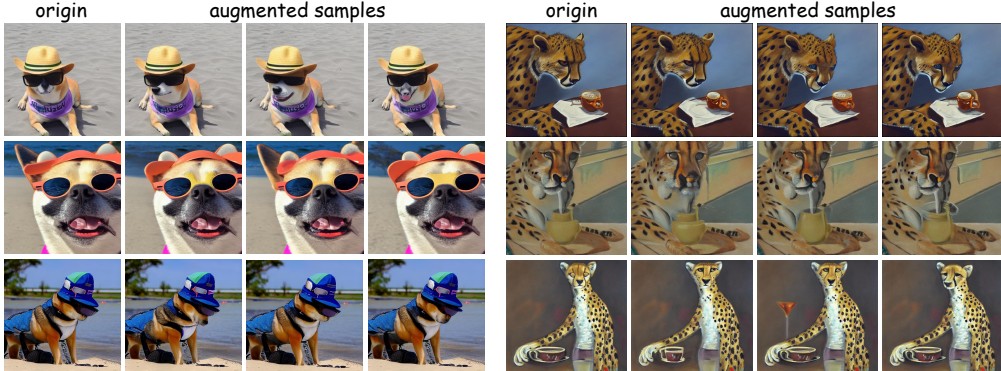

Figure 7: Examples of the augmentation of good generations with enhanced diversity on Stable Diffusion with DDIM sampler (50 NFE).

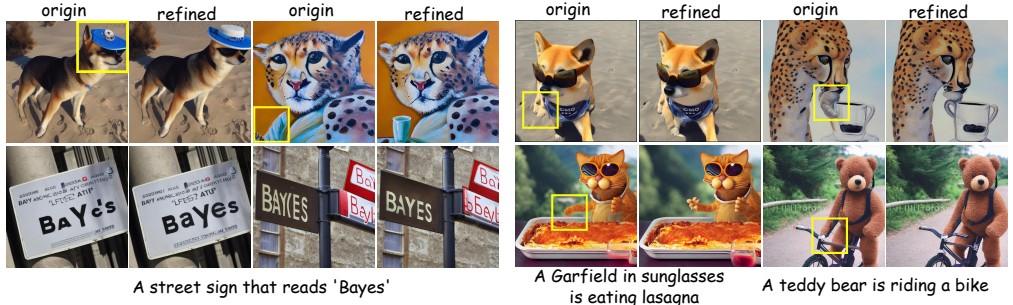

Figure 8: Examples of the rectification of artifacts and misalignment in failure generations on Stable Diffusion with DDIM sampler (50 NFE). The flawed samples are identified by humans and the bounding boxes are manually annotated.

## 4.2 PIXEL-WISE UNCERTAINTY: A TOOL FOR DIVERSITY ENHANCEMENT AND ARTIFACT RECTIFICATION

The main issue with Stable Diffusion is that usually, only a few generations denoised from 'good' initial noises can align with the input textual description (Wu et al., 2023). We show that the pixel-wise uncertainty estimated by BayesDiff can be leveraged to alleviate such an issue. Specifically, BayesDiff-Skip allows for introducing a distribution of $x_t \sim \mathcal{N}(\mathbb{E}(x_t), \mathrm{Var}(x_t))$ for any time $t \in [0, T]$. If the final sample $x_0$ appears to be a good generation, we can resample $x_{t,i}$ from $\mathcal{N}(\mathbb{E}(x_t), \mathrm{Var}(x_t))$ and denoise it to a new one $x_{0,i}$, which is similar yet different from $x_0$. We leverage BayesDiff-Skip with a skipping interval of 1 and DDIM sampler with 50 NFE to test this. We use the Gaussian distribution estimated at $t = 40$ for resampling. Figure 7 shows that the resampled $x_{0,i}$ still conforms the above hypothesis. Moreover, Figure 8 shows that in some cases, the artifacts in original samples can be rectified, and hence the failed samples, which are mismatched with the prompts, are rectified into successful samples. More examples are shown in Appendix A.5.

## 4.3 FURTHER UNDERSTANDING OF PIXEL-WISE UNCERTAINTY

**Visualization of pixel-wise uncertainty.** To gain an intuitive understanding of the obtained uncertainty estimates, we visualize them in this section. In detail, we launch BayesDiff using DDPM model for generating CELEBA images and Stable Diffusion for generating prompt-conditional images. Nonetheless, the visualization for Stable Diffusion is not straightforward because we actually obtain the variance in the final latent states. To solve this problem, we sample a variety of latent states and send them to the decoder of Stable Diffusion. We estimate the empirical variance of the outcomes as the final pixel-wise uncertainty. Figure 9 presents the results. As shown, our uncertainty estimates carry semantic information. The eyes, noses, and mouths of human faces demonstrate greater uncertainty on CELEBA, and the contours of objects in images from Stable Diffusion ex-

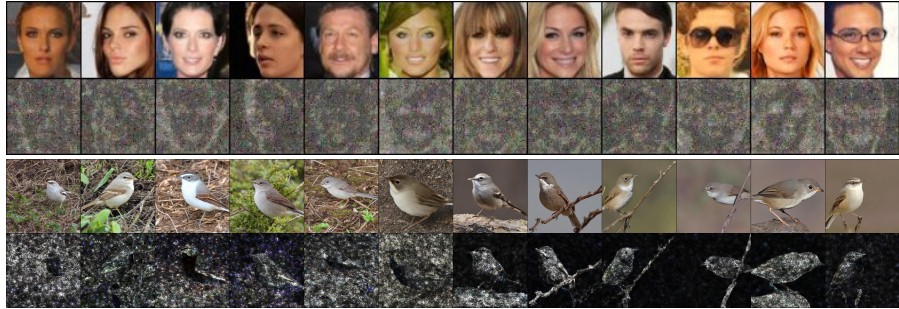

Figure 9: Visualization of the pixel-wise uncertainty of generations on CELEBA (top) and from Stable Diffusion with prompt 'A photo of a whitethroat' (bottom). We adopt BayesDiff-Skip and DDIM sampler (50 NFE) in both cases.

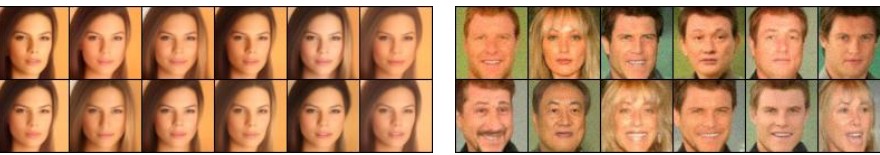

Figure 10: Visualization of $x_{adjacent,0}$, which are denoised from adjacent initial positions corresponding to low image-wise uncertainty (left, ranges from 2.5 to 2.8) and high one (right, ranges from 24.5 to 28.1) on CELEBA with DDIM sampler (50 NFE).

hibit higher levels of uncertainty. This further explains the higher image-wise uncertainty for clutter images and the lower one for clean images mentioned in Section 4.1.

**Visualization of generations of adjacent initial positions.** In BayesDiff, the estimated uncertainty $\mathrm{Var}(x_0)$ is dependent on the corresponding initial position $x_T \sim \mathcal{N}(\mathbf{0}, \mathbf{I})$, and higher uncertainty estimates correspond to larger variations in the sampling trajectory. Therefore, adding minor perturbation to initial positions corresponding to high uncertainty should produce diverse generations. We conduct experiments on CELEBA (Liu et al., 2015) with DDIM sampler to validate this conjecture. Specifically, we define the adjacent initial position $x_{adjacent,T} := \sqrt{1-\eta}x_T + \sqrt{\eta}z, z \sim \mathcal{N}(0, I)$ and visualize $x_{adjacent,0}$, which is denoised from $x_{adjacent,T}$. Figure 10 shows that images generated from adjacent positions with high uncertainty indeed exhibit greater diversity. This result also echos the higher Recall of the group with higher uncertainty on CELEBA in Section 4.1.

## 5 RELATED WORK

Several works incorporate Baysian inference into deep generative models and exihibit strong performance. Variational Autoencoder (VAE) (Kingma & Welling, 2014) is a classic generative model that learns the data distribution through Bayesian variational inference. VAEs have been applied in various domains and demonstrate powerful capabilities of data representation and generation. (Kingma & Welling, 2014; Hou et al., 2017; Bowman et al., 2015; Semeniuta et al., 2017; Wang et al., 2019; Ha & Schmidhuber, 2018) Variational Diffusion Models (VDMs) (Kingma et al., 2021) employ a signal-to-noise ratio function to parameterize the forward noise schedule in diffusion models, enabling the direct optimization of the variational lower bound (VLB) and accelerating training a lot.

## 6 CONCLUSION

In this paper, we introduce BayesDiff, a framework for pixel-wise uncertainty estimation in Diffusion Models via Bayesian inference. We have empirically demonstrated that the estimated pixel-wise uncertainty holds a significant practical value, including being utilized as an image-wise uncertainty metric for filtering low-quality images and a tool for diversity enhancement and misalignment rectification in text-to-image generations. Apart from image generations, the powerful capability of Diffusion Models to generate realistic samples has been applied in various other domains such as natural language processing (audio synthesis (Huang et al., 2022); text-to-speech (Kim et al., 2022)) and AI for science (molecular conformation prediction (Xu et al., 2021); material design (Luo et al., 2022)). We believe BayesDiff holds great potential for incorporating with these applications to improve the predictive uncertainty and calibrate generations.

ETHICS STATEMENT

This work is a fundamental research in machine learning, the potential negative consequences are not apparent. While it is theoretically possible for any technique to be misused, the likelihood of such misuse occurring at the current stage is low.

ACKNOWLEDGMENTS

This work was supported by NSF of China (Nos. 62306176, 62076145), Natural Science Foundation of Shanghai (No. 23ZR1428700), the Key Research and Development Program of Shandong Province, China (No. 2023CXGC010112), and Beijing Outstanding Young Scientist Program (No. BJJWZYJH012019100020098). C. Li was also sponsored by Beijing Nova Program (No. 20220484044).

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

# A APPENDIX

## A.1 ITERATION RULES FOR OTHER SAMPLING METHODS

For Analytic-DPM (Bao et al., 2021), the sampling method is:

$$\mathbf{x}_{t-1} = \frac{1}{\sqrt{\alpha'_t}}(\mathbf{x}_t - \frac{1-\alpha'_t}{\sqrt{1-\bar{\alpha}'_t}}\boldsymbol{\epsilon}_t) + \sigma_t \mathbf{z} \tag{15}$$

$$\sigma_t^2 = \lambda_t^2 + (\sqrt{\frac{1-\bar{\alpha}'_t}{\alpha'_t}} - \sqrt{1-\bar{\alpha}'_{t-1}-\lambda_t^2})^2(1-(1-\bar{\alpha}'_t)\Gamma_t) \tag{16}$$

$$\lambda_t^2 = \frac{1-\bar{\alpha}'_{t-1}}{1-\bar{\alpha}'_t}(1-\alpha'_t), \quad \Gamma_n = \frac{1}{M}\sum_{m=1}^{M}\frac{\|\boldsymbol{s}_t(\boldsymbol{x}_{t,m})\|^2}{d}, \boldsymbol{x}_{t,m} \overset{i.i.d}{\sim} q_t(\boldsymbol{x}_t) \tag{17}$$

The corresponding iteration rule is:

$$\mathbb{E}(\boldsymbol{x}_{t-1}) = \frac{1}{\sqrt{\alpha'_t}}\mathbb{E}(\boldsymbol{x}_t) - \frac{1-\alpha'_t}{\sqrt{\alpha'_t(1-\bar{\alpha}'_t)}}\mathbb{E}(\boldsymbol{\epsilon}_t) \tag{18}$$

$$\mathrm{Var}(\mathbf{x}_{t-1}) = \frac{1}{\alpha'_t}\mathrm{Var}(\mathbf{x}_t) - 2\frac{1-\alpha'_t}{\alpha'_t\sqrt{1-\bar{\alpha}'_t}}\mathrm{Cov}(\boldsymbol{x}_t,\boldsymbol{\epsilon}_t) + \frac{(1-\alpha'_t)^2}{\alpha'_t(1-\bar{\alpha}'_t)}\mathrm{Var}(\boldsymbol{\epsilon}_t) + \sigma_t^2 \tag{19}$$

For DDIM (Song et al., 2020), the sampling method is

$$\boldsymbol{x}_{t-1} = \alpha_{t-1}(\frac{\boldsymbol{x}_t - \sigma_t\boldsymbol{\epsilon}_t}{\alpha_t}) + \sigma_{t-1}\boldsymbol{\epsilon}_t \tag{20}$$

The corresponding iteration rule is:

$$\mathbb{E}(\boldsymbol{x}_{t-1}) = \frac{\alpha_{t-1}}{\alpha_t}\mathbb{E}(\boldsymbol{x}_t) + (\sigma_{t-1} - \frac{\alpha_{t-1}}{\alpha_t}\sigma_t)\mathbb{E}(\boldsymbol{\epsilon}_t) \tag{21}$$

$$\mathrm{Var}(\boldsymbol{x}_{t-1}) = \frac{\alpha_{t-1}^2}{\alpha_t^2}\mathrm{Var}(\boldsymbol{x}_t) + 2\frac{\alpha_{t-1}}{\alpha_t}(\sigma_{t-1} - \frac{\alpha_{t-1}}{\alpha_t}\sigma_t)\mathrm{Cov}(\boldsymbol{x}_t,\boldsymbol{\epsilon}_t) + (\sigma_{t-1} - \frac{\alpha_{t-1}}{\alpha_t}\sigma_t)^2\mathrm{Var}(\boldsymbol{\epsilon}_t) \tag{22}$$

For 2-order DPM-Solver (Lu et al., 2022), applying our method is non-trivial because it involves an extra hidden state between $\boldsymbol{x}_t$ and $\boldsymbol{x}_{t-1}$:

$$\boldsymbol{x}_{s_t} = \frac{\alpha_{s_t}}{\alpha_t}\boldsymbol{x}_t - \sigma_{s_t}(e^{\frac{h_t}{2}}-1)\boldsymbol{\epsilon}_t \tag{23}$$

$$\boldsymbol{x}_{t-1} = \frac{\alpha_{t-1}}{\alpha_t}\boldsymbol{x}_t - \sigma_{t-1}(e^{h_t}-1)\boldsymbol{\epsilon}_{s_t}, \tag{24}$$

where $\lambda_t = \log\frac{\alpha_t}{\sigma_t}$ is the half-log-SNR (Lu et al., 2022). $h_t = \lambda_{t-1} - \lambda_t$. $s_t$ denotes the timestep corresponding to the half-log-SNR of $\frac{\lambda_{t-1}+\lambda_t}{2}$.

Estimating expectation and variance for both sides of Equation (24) yields:

$$\mathbb{E}(\boldsymbol{x}_{t-1}) = \frac{\alpha_{t-1}}{\alpha_t}\mathbb{E}(\boldsymbol{x}_t) - \sigma_{t-1}(e^{h_t}-1)\mathbb{E}(\boldsymbol{\epsilon}_{s_t}) \tag{25}$$

$$\mathrm{Var}(\boldsymbol{x}_{t-1}) = \frac{\alpha_{t-1}^2}{\alpha_t^2}\mathrm{Var}(\boldsymbol{x}_t) - 2\frac{\alpha_{s_t}}{\alpha_t}\sigma_{t-1}(e^{h_t}-1)\mathrm{Cov}(\boldsymbol{x}_t,\boldsymbol{\epsilon}_{s_t}) + \sigma_{t-1}^2(e^{h_t}-1)^2\mathrm{Var}(\boldsymbol{\epsilon}_{s_t}). \tag{26}$$

Unlike first-order sampling methods, we cannot approximate $\mathrm{Cov}(\boldsymbol{x}_t,\boldsymbol{\epsilon}_{s_t})$ with rarely MC samples $\boldsymbol{x}_{t,i}$. Nonetheless, we observe that injecting the uncertainty on $\boldsymbol{\epsilon}_t$ to Equation (23) yields

$$\boldsymbol{x}_{s_t}|\boldsymbol{x}_t \sim \mathcal{N}(\frac{\alpha_{s_t}}{\alpha_t}\boldsymbol{x}_t - \sigma_{s_t}(e^{\frac{h_t}{2}}-1)\boldsymbol{\epsilon}_\theta(\boldsymbol{x}_t,t), \mathrm{diag}(\sigma_{s_t}^2(e^{\frac{h_t}{2}}-1)^2\gamma_\theta^2(\boldsymbol{x}_t,t))). \tag{27}$$

Consequently, we can first sample $\boldsymbol{x}_{t,i} \sim \mathcal{N}(\mathbb{E}(\boldsymbol{x}_t), \mathrm{Var}(\boldsymbol{x}_t))$, based on which $\boldsymbol{x}_{s_t,i}$ are sampled. Then, we can approximate $\mathrm{Cov}(\boldsymbol{x}_t,\boldsymbol{\epsilon}_{s_t})$ with MC estimation similar to Equation (11).

## A.2 UNCERTAINTY QUANTIFICATION ON CONTINUOUS-TIME REVERSE PROCESS

Firstly, we integrate Equation (6) with time $t$ from 0 to $T$ to obtain the continuous-time solution $\boldsymbol{x}_0$.

$$x_0 = x_T - \int_0^T [f(t)\boldsymbol{x}_t + \frac{g(t)^2}{\sigma_t}\boldsymbol{\epsilon}_t]dt + \int_0^T g(t)d\bar{\boldsymbol{\omega}}_t \tag{28}$$

Due to the independence between the reverse-time Wiener process and $x_t$, we have

$$
\begin{aligned}
\mathrm{Var}(\boldsymbol{x}_0) &= \mathrm{Var}(\boldsymbol{x}_T) + \mathrm{Var}(\int_0^T [f(t)\boldsymbol{x}_t + \frac{g(t)^2}{\sigma_t}\boldsymbol{\epsilon}_t]dt) + \mathrm{Var}(\int_0^T g(t)d\bar{\boldsymbol{\omega}}_t) \\
&= \mathbf{1} + \mathrm{Var}(\int_0^T [f(t)\boldsymbol{x}_t + \frac{g(t)^2}{\sigma_t}\boldsymbol{\epsilon}_t]dt) + \int_0^T g(t)^2 dt
\end{aligned}
\tag{29}
$$

The second equality is derived using the Ito isometry property Equation (30) of Ito calculus:

$$\mathbb{E}(\int_0^T g_1(t)d\bar{\boldsymbol{\omega}}_t \int_0^T g_2(t)d\bar{\boldsymbol{\omega}}_t) = \mathbb{E}(\int_0^T g_1(t)g_2(t)dt) \tag{30}$$

that is,

$$
\begin{aligned}
\mathrm{Var}(\int_0^T g(t)d\bar{\boldsymbol{\omega}}_t) &= \mathbb{E}((\int_0^T g(t)d\bar{\boldsymbol{\omega}}_t)^2) - \mathbb{E}(\int_0^T g(t)d\bar{\boldsymbol{\omega}}_t)^2 \\
&= \mathbb{E}((\int_0^T g(t)d\bar{\boldsymbol{\omega}}_t)^2) - 0 = \int_0^T g(t)^2 dt
\end{aligned}
\tag{31}
$$

Assuming that the reward process $\boldsymbol{x}_t, t \in [0, T]$ is a stochastic process with second order moments and is mean square integrable, we have

$$
\begin{aligned}
&\mathrm{Var}(\int_0^T [f(t)\boldsymbol{x}_t + \frac{g(t)^2}{\sigma_t}\boldsymbol{\epsilon}_t]dt) \\
&= \int_0^T \int_0^T f(s)f(t)\mathrm{Cov}(\boldsymbol{x}_s, \boldsymbol{x}_t) \\
&\quad -f(s)\frac{g(t)^2}{\sigma_t}\mathrm{Cov}(\boldsymbol{x}_s, \boldsymbol{\epsilon}_t) - f(t)\frac{g(s)^2}{\sigma_s}\mathrm{Cov}(\boldsymbol{x}_t, \boldsymbol{\epsilon}_s) + \frac{g(s)^2}{\sigma_s}\frac{g(t)^2}{\sigma_t}\mathrm{Cov}(\boldsymbol{\epsilon}_s, \boldsymbol{\epsilon}_t)dsdt \\
&= \int_0^T \int_0^T f(s)f(t)\mathrm{Cov}(\boldsymbol{x}_s, \boldsymbol{x}_t) - f(s)\frac{g(t)^2}{\sigma_t}\mathrm{Cov}(\boldsymbol{x}_s, \boldsymbol{\epsilon}_t)dsdt \\
&\quad -2\int_0^T f(s)\frac{g(s)^2}{\sigma_s}\mathrm{Cov}(\boldsymbol{x}_s, \boldsymbol{\epsilon}_s)ds + \int_0^T \frac{g(s)^4}{\sigma_s^2}\mathrm{Cov}(\boldsymbol{\epsilon}_s, \boldsymbol{\epsilon}_s)ds
\end{aligned}
\tag{32}
$$

The second equvalence holds because $\boldsymbol{x}_s$ and $\boldsymbol{\epsilon}_t$, $\boldsymbol{\epsilon}_s$ and $\boldsymbol{\epsilon}_t$ are both independent when $s \neq t$, i.e. $\mathrm{Cov}(\boldsymbol{x}_s, \boldsymbol{\epsilon}_t) = 0, \mathrm{Cov}(\boldsymbol{\epsilon}_s, \boldsymbol{\epsilon}_t) = 0, \forall s \neq t$.

$$
\begin{aligned}
\int_0^T \int_0^T f(s)f(t)\mathrm{Cov}(\boldsymbol{x}_s, \boldsymbol{x}_t)dsdt &= 2\int_0^T f(t)(\int_0^t f(s)\mathrm{Cov}(\boldsymbol{x}_s, \boldsymbol{x}_t)ds)dt \\
&= 2\int_0^T f(t)(\int_0^t f(s)\mathrm{Cov}(\boldsymbol{x}_s, \boldsymbol{x}_s + \int_s^t (f(u)\boldsymbol{x}_u - \frac{g(u)^2}{\sigma_u}\boldsymbol{\epsilon}_u)du \\
&\quad + \int_s^t g(u)d\bar{\boldsymbol{\omega}}_u ds)dt \\
&= 2\int_0^T f(t)(\int_0^t f(s)\mathrm{Var}(\boldsymbol{x}_s)ds)dt \\
&\quad + 2\int_0^T f(t)(\int_0^t f(s)\mathrm{Cov}(\boldsymbol{x}_s, \int_s^t (f(u)\boldsymbol{x}_u - \frac{g(u)^2}{\sigma_u}\boldsymbol{\epsilon}_u)du)ds)dt
\end{aligned}
\tag{33}
$$

Then the question boils down to approximate $\text{Cov}(\boldsymbol{x}_s, \int_s^t (f(u)\boldsymbol{x}_u - \frac{g(u)^2}{\sigma_u}\boldsymbol{\epsilon}_u)du)$. According to the numerical integration method and $\text{Cov}(\boldsymbol{x}_s, \boldsymbol{\epsilon}_t) = 0, \forall s \neq t$,

$$
\begin{aligned}
\text{Cov}(\boldsymbol{x}_s, \int_s^t (f(u)\boldsymbol{x}_u - \frac{g(u)^2}{\sigma_u}\boldsymbol{\epsilon}_u)du) &= \text{Cov}(\boldsymbol{x}_s, \int_s^t f(u)\boldsymbol{x}_u du) + \text{Cov}(\boldsymbol{x}_s, \int_s^t -\frac{g(u)^2}{\sigma_u}\boldsymbol{\epsilon}_u)du) \\
&= \text{Cov}(\boldsymbol{x}_s, \int_s^t f(u)\boldsymbol{x}_u du) - \frac{g(u)^2}{\sigma_u}\text{Cov}(\boldsymbol{x}_s, \boldsymbol{\epsilon}_u)\Delta t
\end{aligned}
\tag{34}
$$

So when $\Delta t \to 0$,

$$
\text{Cov}(\boldsymbol{x}_s, \int_s^t (f(u)\boldsymbol{x}_u - \frac{g(u)^2}{\sigma_u}\boldsymbol{\epsilon}_u)du) = \text{Cov}(\boldsymbol{x}_s, \int_s^t f(u)\boldsymbol{x}_u du)
\tag{35}
$$

According to sampling method $\boldsymbol{x}_{t-1} = \boldsymbol{x}_t - (f(t)\boldsymbol{x}_t + \frac{g(t)^2}{\sigma_t}\boldsymbol{\epsilon}_t))\Delta t + g(t)(\boldsymbol{w}_{t-1} - \boldsymbol{w}_t)$,

$$
x_{s+\Delta t} = x_s + (f(s)\boldsymbol{x}_s + \frac{g(s)^2}{\sigma_s}\boldsymbol{\epsilon}_s)\Delta t + h(w)
$$

$$
x_{s+2\Delta t} = x_{s+\Delta t} + (f(s+\Delta t)\boldsymbol{x}_{s+\Delta t} + \frac{g(s+\Delta t)^2}{\sigma_{t+\Delta t}}\boldsymbol{\epsilon}_{s+\Delta t})\Delta t + h(w)
$$

$$
= x_s + (f(s)\boldsymbol{x}_s + \frac{g(s)^2}{\sigma_s}\boldsymbol{\epsilon}_s)\Delta t + f(s+\Delta t)\boldsymbol{x}_s\Delta t + \frac{g(s+\Delta t)^2}{\sigma_{t+\Delta t}}\boldsymbol{\epsilon}_{s+\Delta t}\Delta t + \mathcal{O}(\Delta t^2) + h(w)
\tag{36}
$$

Then using naiive numerical integration method, we have

$$
\begin{aligned}
\int_s^t f(u)\boldsymbol{x}_u du &= f(s)\boldsymbol{x}_s\Delta t + f(s+\Delta t)\boldsymbol{x}_{s+\Delta t}\Delta t + f(s+2\Delta t)\boldsymbol{x}_{s+2\Delta t}\Delta t + \cdots + + f(t-\Delta t)\boldsymbol{x}_{t-\Delta t}\Delta t \\
&= f(s)\boldsymbol{x}_s\Delta t + f(s+\Delta t)\boldsymbol{x}_s\Delta t + f(s+2\Delta t)\boldsymbol{x}_s\Delta t + \cdots + f(t-\Delta t)\boldsymbol{x}_s\Delta t + \sum \mathcal{O}(\Delta t^2) + H(w) \\
&= \int_s^t f(u)\boldsymbol{x}_s du + \sum \mathcal{O}(\Delta t^2) + H(w)
\end{aligned}
\tag{37}
$$

We neglect second-order terms and get the approximation of $\text{Cov}(\boldsymbol{x}_s, \int_s^t f(u)\boldsymbol{x}_u du)$:

$$
\text{Cov}(\boldsymbol{x}_s, \int_s^t f(u)\boldsymbol{x}_u du) \approx \text{Cov}(\boldsymbol{x}_s, \int_s^t f(u)\boldsymbol{x}_s du) = \text{Var}(\boldsymbol{x}_s)\int_s^t f(u)du
\tag{38}
$$

In conclusion, we derive an approximate illustrating the pattern of uncertainty dynamics from $\boldsymbol{x}_T$ to $\boldsymbol{x}_0$,

$$
\begin{aligned}
\text{Var}(\boldsymbol{x}_0) \approx \mathbf{1} &+ 2\int_0^T f(t)(\int_0^t f(s)\text{Var}(\boldsymbol{x}_s)ds)dt + 2\int_0^T f(t)(\int_0^t f(s)\text{Var}(\boldsymbol{x}_s)(\int_s^t f(u)du)ds)dt \\
&- 2\int_0^T f(s)\frac{g(s)^2}{\sigma_s}\text{Cov}(\boldsymbol{x}_s, \boldsymbol{\epsilon}_s)ds + \int_0^T \frac{g(s)^4}{\sigma_s^2}\text{Var}(\boldsymbol{\epsilon}_s)ds + \int_0^T g(t)^2 dt
\end{aligned}
\tag{39}
$$

Moreover, we can generalize it to arbitrary reverse time interval $[i, j] \in [0, T]$. Specifically, $\forall 0 \leq i \leq j \leq T$,

$$
\begin{aligned}
\text{Var}(\boldsymbol{x}_i) \approx \text{Var}(\boldsymbol{x}_j) &+ 2\int_i^j f(t)(\int_i^t f(s)\text{Var}(\boldsymbol{x}_s)ds)dt + 2\int_i^j f(t)(\int_i^t f(s)\text{Var}(\boldsymbol{x}_s)(\int_s^t f(u)du)ds)dt \\
&- 2\int_i^j f(s)\frac{g(s)^2}{\sigma_s}\text{Cov}(\boldsymbol{x}_s, \boldsymbol{\epsilon}_s)ds + \int_i^j \frac{g(s)^4}{\sigma_s^2}\text{Var}(\boldsymbol{\epsilon}_s)ds + \int_i^j g(t)^2 dt
\end{aligned}
\tag{40}
$$

---

**Algorithm 2** A faster variant of BayesDiff. (BayesDiff-Skip)

---

**Input:** Starting point $\boldsymbol{x}_T$, Monte Carlo sample size $S$, Pre-trained noise prediction model $\epsilon_\theta$.
**Output:** Image generation $\boldsymbol{x}_0$ and pixel-wise uncertainty $\mathrm{Var}(\boldsymbol{x}_0)$.
 1: Construct the pixel-wise variance prediction function $\gamma_\theta^2$ via LLLA;
 2: $\mathbb{E}(\boldsymbol{x}_T) \leftarrow \boldsymbol{x}_T, \mathrm{Var}(\boldsymbol{x}_T) \leftarrow \boldsymbol{0}, \mathrm{Cov}(\boldsymbol{x}_T, \boldsymbol{\epsilon}_T) \leftarrow \boldsymbol{0}$;
 3: **for** $t = T \rightarrow 1$ **do**
 4:     **if** $t \in \tilde{\boldsymbol{t}}$ **then**
 5:         Sample $\boldsymbol{\epsilon}_t \sim \mathcal{N}(\epsilon_\theta(\boldsymbol{x}_t, t), \mathrm{diag}(\gamma_\theta^2(\boldsymbol{x}_t, t)))$;
 6:     **else**
 7:         $\boldsymbol{\epsilon}_t \leftarrow \epsilon_\theta(\boldsymbol{x}_t, t), \mathrm{Cov}(\boldsymbol{x}_t, \boldsymbol{\epsilon}_t) \leftarrow \boldsymbol{0}, \mathrm{Var}(\boldsymbol{\epsilon}_t) \leftarrow \boldsymbol{0}$;
 8:     **end if**
 9:     Obtain $\boldsymbol{x}_{t-1}$ via Equation (7);
10:     Estimate $\mathbb{E}(\boldsymbol{x}_{t-1})$ and $\mathrm{Var}(\boldsymbol{x}_{t-1})$ via Equation (10) and Equation (8);
11:     **if** $t - 1 \in \tilde{\boldsymbol{t}}$ **then**
12:         Sample $\boldsymbol{x}_{t-1,i} \sim \mathcal{N}(\mathbb{E}(\boldsymbol{x}_{t-1}), \mathrm{Var}(\boldsymbol{x}_{t-1})), i = 1, \ldots, S$;
13:         Estimate $\mathrm{Cov}(\boldsymbol{x}_{t-1}, \boldsymbol{\epsilon}_{t-1})$ via Equation (11);
14:     **end if**
15: **end for**

---

### A.3 BAYESDIFF-SKIP ALOGORITHM

In this section, we present our BayesDiff-Skip algorithm. To be specific, if sampling with uncertainty is used from $\boldsymbol{x}_t$ to $\boldsymbol{x}_{t-1}$, then $\boldsymbol{\epsilon}_t$ is considered as a random variable sampled from the normal posterior predictive distribution, where $\mathrm{Cov}(\boldsymbol{x}_t, \boldsymbol{\epsilon}_t)$ and $\mathrm{Var}(\boldsymbol{\epsilon}_t)$ are non-zero. Conversely, if original deterministic sampling is used from $\boldsymbol{x}_t$ to $\boldsymbol{x}_{t-1}$, $\boldsymbol{\epsilon}_t$ is treated as a constant and $\mathrm{Cov}(\boldsymbol{x}_t, \boldsymbol{\epsilon}_t)$ and $\mathrm{Var}(\boldsymbol{\epsilon}_t)$ are zero. We conclude it to this algorithm in Algorithm 2.

### A.4 IMPLEMENTATION DETAILS OF LAST LAYER LAPLACE APPROXIMATION

We adopt the most lightweight diagonal factorization and ignore off-diagonal elements for Hessian approximation in LLLA (Daxberger et al., 2021a). To avoid storing large Jacobian matrix, we adopt the Monte Carlo method to approximate the accurate variance of outputs $diag(\gamma_\theta^2(x, t))$ directly by the variance of samples $f_{\theta_i}(x, t), \theta_i \sim p(\theta|\mathcal{D}) = \mathcal{N}(\theta; \theta_{\mathrm{MAP}}, \boldsymbol{\Sigma})$. This results in faster computation speed, while still maintaining a reasonable level of accuracy. The number of samples is chosen as 100 in practice.

### A.5 ADDITIONAL EXAMPLES OF RESAMPLING METHOD WITH BAYESDIFF

To provide a more intuitive demonstration of the significance of our resampling method, we include additional prompts and corresponding images containing artifacts annotated by humans, presenting 8 resampled images using BayesDiff in Figure 11 and Figure 12. We find that the success rate of resampling flawed samples into desirable samples is approximately at least $60\%$.

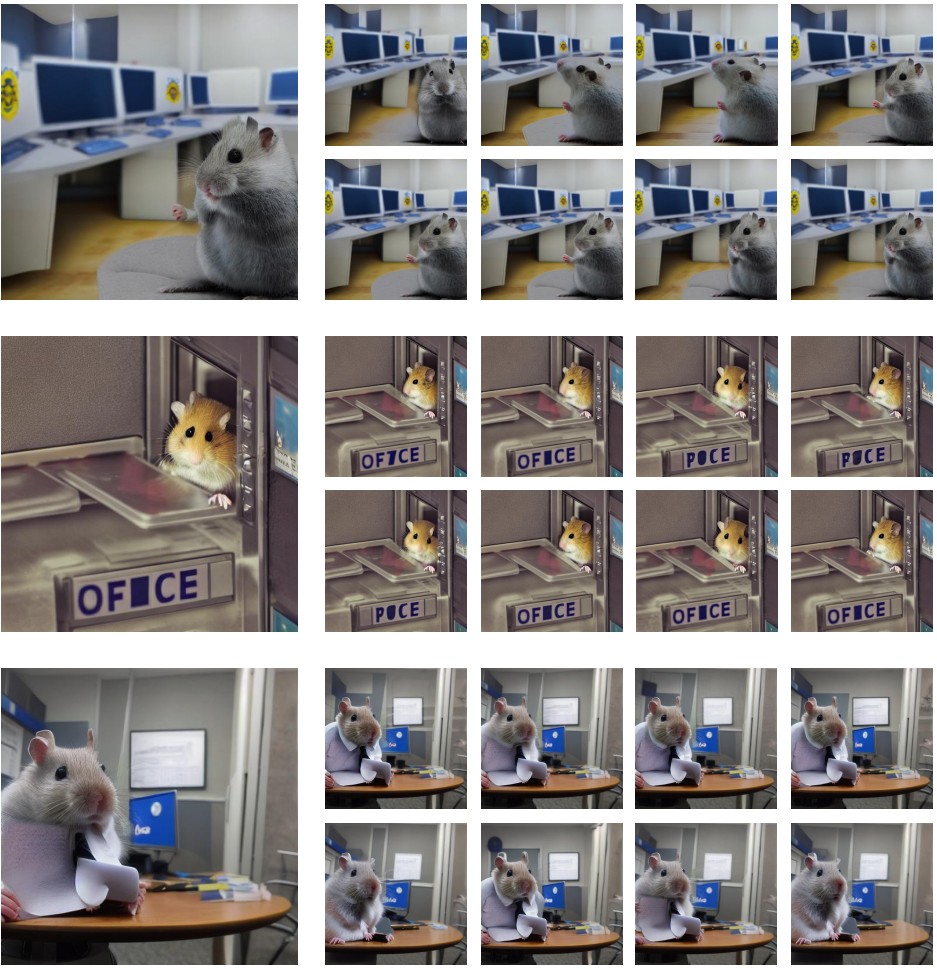

a hamster working in a police office, professional photography, photo realistic

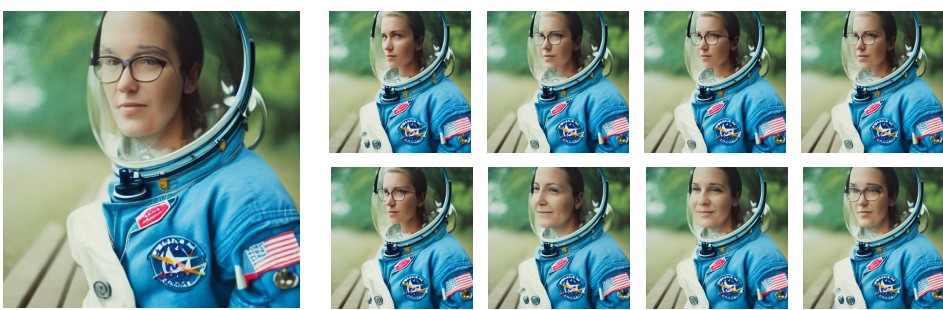

medium format film portrait close up of a female astronaut sitting on a bench in the park on a rainy day, hasselblad film bokeh, unsplash, soft light photographed on colour expired film

Figure 11: 8 potential rectified samples resampled by BayesDiff (right) of flawed images (left) annotated by humans.

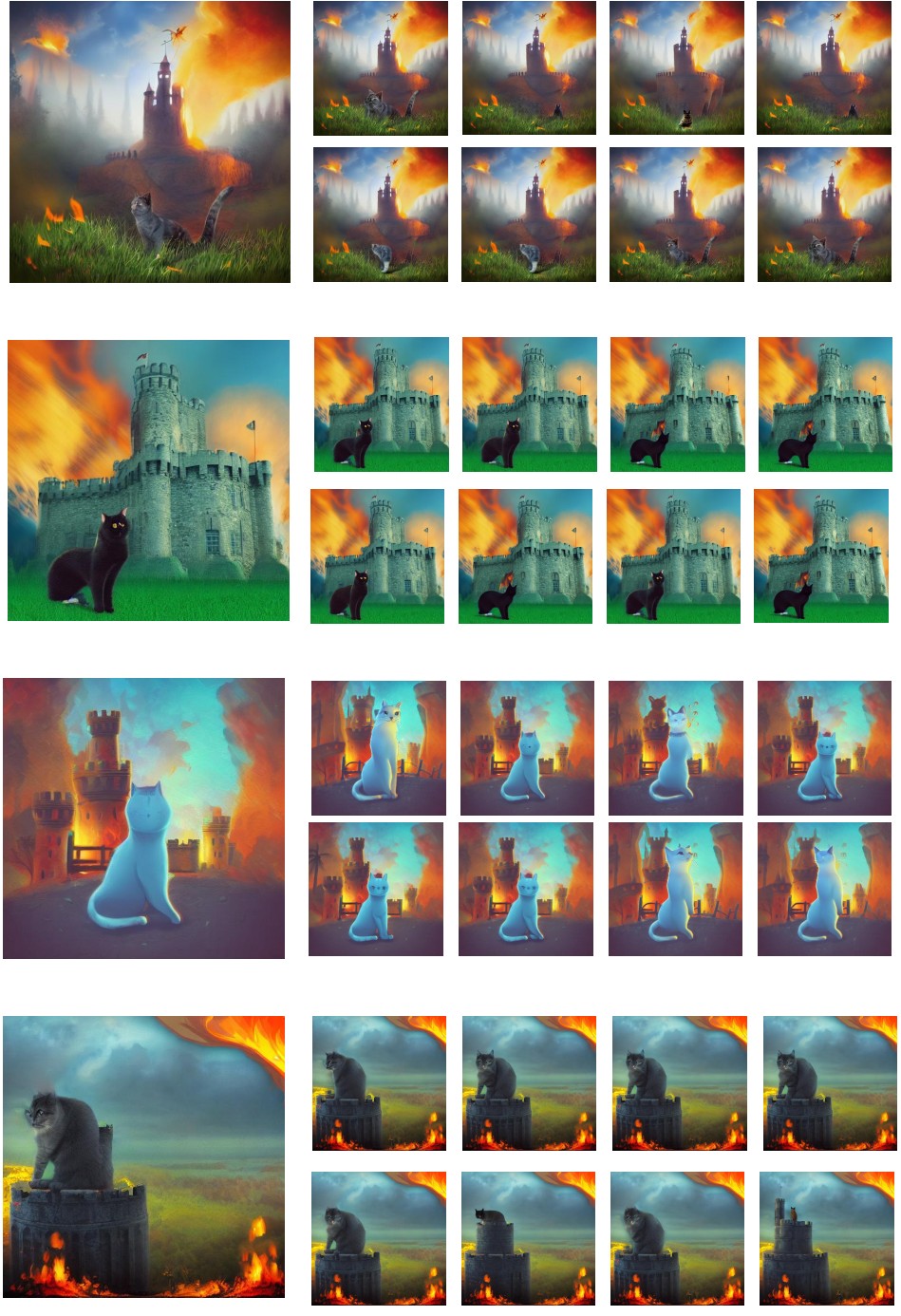

a cat standing on a castle surrounded by fire, digital art, realistic, large
depth of field, vignette effect, trending on arstation

Figure 12: 8 potential rectified samples resampled by BayesDiff (right) of flawed images (left) annotated by humans.

