# OpenReview forum: "BayesDiff: Estimating Pixel-wise Uncertainty in Diffusion via Bayesian Inference"
_ICLR.cc/2024/Conference — ICLR 2024 poster_

### Official Review · Reviewer_AU5E · 2023-10-15

**Soundness:** 3 good
**Presentation:** 2 fair
**Contribution:** 3 good
**Rating:** 8
**Confidence:** 3

**Summary:**

This paper proposes to leverage epistemic uncertainty in the (reverse) sampling process of diffusion models. This is done by applying the last-layer Laplace approximation to the image-to-image neural network that approximates the score function. Using some approximations, this uncertainty gives rise to uncertainty in the image sample at each time step of the backward diffusion process. Applications like filtering low-quality samples and sample diversity enhancement are discussed.

**Strengths:**

I find the paper is well-written and easy to follow even for me, who is not very familiar with diffusion models. In any case, the proposed method is sound and most importantly, very practical---the authors noted that the overhead of their method is no more than 1x of the standard diffusion sampling.

I especially find the applications presented to be very interesting, illuminating, and again: practical. I emphasize the practicality of this paper since Bayesian neural networks are often quite impractical for large-scale problems like diffusion models.

**Weaknesses:**

1. Some details that might be useful for potential readers are glossed over. E.g. how is $\mathrm{diag}(\gamma_\theta^2(x_t, t))$ computed? What approximation of the Hessian is used? etc.
2. Sec. 4.2 and 4.3 are a bit handwavy---it would be much better if the authors could make them more quantitative, like Sec. 4.1. Handpicked examples are not useful to instill confidence about the benefits of BayesDiff.
3. Some figures are quite hard to follow:
    1. Fig. 2 is quite hard to understand. The caption is not descriptive at all and the colors are hard to see in print.
    2. Fig. 3 & 4: need more spacing between "left" and "right" groups. I was really confused at first trying to parse what is "left" and what is "right".
    3. Fig. 5: The colors are horrible (esp. in print), they're indistinguishable. It's better to use different markers or different linestyle instead.

**Questions:**

The last-layer Laplace approximation can still be very expensive for networks with high output dimensionality, e.g. in text or image generation. For example, if the image is $d \times d$, then the network has an output dim of $d^2$. Assuming the last-layer feature dim of $h$, this means the last-layer weight matrix is $d^2 \times h$ and so the Hessian is $hd^2 \times hd^2$. Then, to get the variance over outputs $\gamma^2(x, t)$, you need to multiply the Hessian with the last-layer Jacobian, which itself is large---$d^2 \times hd^2$. Can the authors elaborate on how BayesDiff overcomes this issue in practice?

---

> ### Author Response · Authors · 2023-11-16
> **Response to Reviewer AU5E**
>
> Thank you for the positive feedback and useful suggestions! We are glad you noted the practical benefits. We address your concerns point by point below.
>
> W1 & Q: Implementation details of the Laplace approximation.
>
> Thank you for pointing out that we lack implementation details regarding the Laplace approximation, we have added this to our paper.
>
> We adopt the most lightweight diagonal factorization and ignore off-diagonal elements for Hessian approximation [1]. Regarding the covariance $\gamma_\theta^2(x, t) \in R^{h d^2 * h d^2}$ of the posterior predictive distribution, as you point out, to get the accurate covariance of outputs is indeed time-consuming (17s when the image size is 64*64). Luckily, in our settings, we do not need to estimate the entire covariance matrix; we only need to estimate the diagonal. So we adopt the Monte Carlo method to approximate the accurate variance of outputs $diag(\gamma_\theta^2(x, t))$ directly by the variance of samples { $ f_{\theta_i}(x,t) $ }, $\theta_i  \sim p(\theta|\mathcal{D}) =\mathcal{N}(\theta ; \theta_{\text{MAP}}$, $\boldsymbol{\Sigma})$. This avoids storing large Jacobian matrix, resulting in faster computation speed, while still maintaining a reasonable level of accuracy.
>
> The number of samples is chosen as 100 in practice. That is, by performing 99 additional inferences of the last layer, we can accurately estimate the posterior distribution without significant computational or time constraints (the computational time decreases from 17s to 0.1s). Furthermore, when comparing the results obtained from the accurate variance and the approximated variance, the average approximation error per pixel is 10%. It is minor and can be negligible because the magnitude of the variance relative to the mean is significantly small.
>
> W2: Qualitative nature of the results presented in Section 4.2.
>
> For flawed samples exhibiting artifacts and misalignments selected by humans, we can provide possibilities for generating potential refined samples with BayesDiff. However, it is important to note that quantitatively measuring such value on a large scale poses challenges as there are no benchmarks or quantitative metrics to assess the presence of high-granularity artifacts. Therefore, the effectiveness of our method can only be demonstrated through the case study based on human evaluations. To provide a more intuitive demonstration of the significance of our resampling method, we have included additional prompts and corresponding images containing artifacts annotated by humans in Appendix A.7. By observing the resampling results, we find that the success rate of resampling flawed samples into desirable samples is approximately at least 60%. We will involve more human experts in the evaluation process in the final version.
>
> W3: Suggestions on figures in the paper.
>
> Thanks for your useful suggestions! We have made the modifications to the figures based on your suggestions. If you have any further advice, please feel free to provide them.
>
> Given these improvements and changes to your initial concerns, we hope for your reconsideration in raising your score.
>
> [1] Daxberger E, Kristiadi A, Immer A, et al. Laplace redux-effortless bayesian deep learning. Advances in Neural Information Processing Systems 2021.

---

> > ### Comment · Reviewer_AU5E · 2023-11-20
> > **Reply**
> >
> > Thanks for your clarification! I don't have further questions. I updated my score accordingly.

---

> ### Author Response · Authors · 2023-11-21
> **Acknowledgement**
>
> Thank you again for your valuable feedback on our submission.

---

### Official Review · Reviewer_CcvE · 2023-10-25

**Soundness:** 3 good
**Presentation:** 3 good
**Contribution:** 2 fair
**Rating:** 6
**Confidence:** 2

**Summary:**

This paper proposes to use a last-layer Laplace approximation in diffusion models and derives how to propagate variance iteratively through the diffusion dynamics to obtain per-pixel uncertainty estimates. The experiments leverage these for filtering out low-fidelity samples, rectifying visual artifacts and visualization.

Overall the method seems sensible, although I am unfortunately not too familiar with diffusion models, so may not be the best person to judge this. The evaluation seems to largely rely on subjective, qualitative analysis, and where it is quantitative the differences are mostly small and error bars missing. So all in all I would slightly lean towards rejection, but I am not strongly opinionated either way due to a lack of confidence.

EDIT: In light of the rebuttal, I now lean towards acceptance.

**Strengths:**

* Overall the methodology seems sensible, utilizing the fact that diffusion models are probabilisitic to perform inference is quite a natural approach.
* The paper is well structured and clear on what prior work it builds on.

**Weaknesses:**

* I find it hard to tell whether the proposed method does anything meaningful. Most of the comparisons are rather qualitative, and where they are quantitative they are hard to interpret, e.g. it is quite difficult to decide what to make of Table 1. Many of the differences are quite small and without error bars it seems impossible to know whether those correspond to meaningful performance gains.
* There are no baselines. I appreciate that there may not have been any prior work in this direction (although I would imagine that there would be some non-probabilistic filtering techniques. Perhaps from the literature on GANs?), however given that the iterative sampling process involves a Gaussian at every step, if I am understanding things correctly I would think that a deterministic diffusion model would also give us pixelwise variances that could be used as a baseline.
* Alternatively, it might have been interesting to experiment with different covariance structures for the Laplace approximation to see if those make a difference.
* I did not find the background section on diffusion models (2.1) particularly helpful as it relies on a lot of terminology on SDEs. The opening paragraph is good, perhaps something similar that briefly summarizes things from an algorithmic perspective (what kind of network are we typically training to predict what and w.r.t. what objective, what is being sampled, ...).

**Questions:**

* I would like to see error bars for (some of) the quantitative results.
* Is there a strict need to use a Laplace approximation for inference? The likelihood is a Gaussian, so if you are only estimating uncertainty over the final layer weights, shouldn't the posterior be Gaussian as well? Or is this not the case due to the iterative sampling process?
* Could the sampling variances be used to create a baseline with a deterministic diffusion model?

Minor:
* I think it would be helpful to complement Figure 2 with a plot of skipping intervals vs Spearman correlation with the no-skipping ranking.
* For Figure 5, I would suggest using a different color palette (with different colors rather than differing shades) and distinct markers. It is unnecessarily difficult to match the lines and legends as is.

---

> ### Author Response · Authors · 2023-11-16
> **Response to Reviewer CcvE**
>
> Thank you for your attentive comments and useful suggestions! We are glad you thought our method was sensible. We address your feedback point by point below.
>
> W1 & Q1: Error bar of results in Table 1.
>
> Table 1 showcases a comparison of metrics between randomly selected 50,000 images and the remaining 50,000 images after filtering out those with the highest uncertainty from 60,000 generated images. In response to your feedback, we conducted multiple random selections (10 different random seeds) of 50,000 images and provided error bars for the FID metric of the randomly selected images. **Results added in Table 1 show that the improvement in the overall distribution metrics by filtering out low-quality images based on the sample-wise uncertainty metric is significant compared to the variance of the measurement.**
>
> Moreover,  we would like to reiterate the significance of our method. There are still low-quality generations in large-scale models currently, and it is challenging to identify such individual images using traditional metrics (FID, IS, LIPSIS) that measure the quality of a large set of images distribution-wisely. The core value of our approach lies in introducing a sample-wise metric based on pixel-wise uncertainty. This sample-wise metric enables us to effectively filter out cluttered low-quality images.
>
> Q2：Is there a strict need to use a Laplace approximation for inference?
>
> Sorry that we did not explain this perfectly. You are correct that the posterior distribution of the parameters is indeed a Gaussian distribution in our settings. However, conventional methods require a significant amount of data to fit the Gaussian posterior distribution, which can be computationally expensive. **The reason we opt for the Laplace approximation is to directly leverage the parameters of the pre-trained model.** Under this approximation, the parameters of the pre-trained model can serve as the Maximum A-Posteriori (MAP) estimate of the Gaussian posterior distribution, i.e. the mean of the Gaussian distribution is approximated by the pre-trained weights. It is training-free and facilitates practical applications.
>
> W2 & Q3: Could the sampling variances be used to create a baseline with a deterministic diffusion model?
>
> Thank you for your appreciation of our work as the first one in this direction. Regarding the baseline, as far as I know, most filtering techniques do not apply to our task because they focus on detecting deepfake images or filtering out low-quality images from training datasets. [1, 2] There is currently no widely accepted sample-wise metric that can be used to detect low-quality images generated by large-scale generative models, so we adopt the randomly filtered data as a comparison in our quantitative result. On the other hand, **the sampling variances with a deterministic diffusion model cannot be used to create a baseline.** This is because, for an ODE solver-based reverse generative process, the sampling process does not follow a Gaussian distribution but is deterministic, lacking sample variance. As for an SDE solver-based reverse generative process, all samples' sampling variances are equal, making it unable to distinguish between samples.
>
> W3: Other approximation methods for Hessian.
> Indeed, our method supports the use of other approximation methods for Hessian, although that is not the primary focus of our contribution. The experimental section primarily emphasizes the understanding and utilization of pixel-wise variance. However, your point about exploring alternative approximation methods is intriguing, and we will explore it as part of our future work.
>
> W4: Suggestions on Section 2.1.
>
> We have incorporated the common architectural selection of the neural network and specific sampling methods in Section 2.1 based on your suggestions. If you have any further advice, please feel free to share.
>
> Given these improvements and changes to your initial concerns, we hope you would like to raise your score.
>
> [1] Le B M, Woo S S. Add: Frequency attention and multi-view based knowledge distillation to detect low-quality compressed deepfake images. AAAI 2022.
>
> [2] Wong Y, Chen S, Mau S, et al. Patch-based probabilistic image quality assessment for face selection and improved video-based face recognition. CVPR 2011 WORKSHOPS.

---

> > ### Author Response · Authors · 2023-11-21
> > **Sincerely looking forward to the further discussions**
> >
> > Dear reviewer,
> >
> > We are wondering if our response and revision have resolved your concerns. If our response has addressed your concerns, we would highly appreciate it if you could re-evaluate our work and consider raising the score.
> >
> > If you have any additional questions or suggestions, we would be happy to have further discussions.
> >
> > Best regards,
> >
> > The Authors

---

> > > ### Comment · Reviewer_CcvE · 2023-11-21
> > >
> > > Thank you for the clarifications and updates, I will increase my score.

---

> ### Author Response · Authors · 2023-11-22
> **Acknowledgement**
>
> Thank you for your feedback！

---

### Official Review · Reviewer_NYDr · 2023-10-30

**Soundness:** 3 good
**Presentation:** 2 fair
**Contribution:** 2 fair
**Rating:** 6
**Confidence:** 4

**Summary:**

Diffusion models are powerful generative models but it is not easy to output standard bayesian uncertainty statistics from them such as posterior predictive probability, or pixel-wise uncertainty etc. Knowing if there are some pixels in an image, or an entire image, can be very helpful in ensuring high quality in downstream tasks.

This work uses the well known method of Laplacian approximation to estimate parameter uncertainty in the score network of an image diffusion model. For computational efficiency, and tractability in converting parameter uncertainty into sample uncertainty the authors use the well known approximation to only estimate the variance in the last linear layer of the neural network. Once the uncertainty update from a single application of the score network can be computed then the final uncertainty can be easily computed by deriving the update equation for different first-order and second order discrete time samplers.

==== After rebuttal ======
Thanks for the updates. No changes to rating. Best wishes.

**Strengths:**

The paper is reasonably novel and presents a methodology that practitioners may find useful. Specially the use of pixel-uncertainty in fixing the sample may be useful.

**Weaknesses:**

While the overall method is quite simple and the experiments show some potential for the method but the actual experiments are not clear/substantive enough. See questions section for more details.

**Questions:**

1. Section 4.2 shows that pixel wise uncertainty can be used to correct bad portions / artefacts in the original images. Many questions come to mind about this experiment. Were the bounding boxes for the artefacts determined automatically based on pixel uncertainty ? Even if they were identified manually ? How  often are the refined samples sampled using rejection sampling on pixel uncertainty score better than the original ? In other words are the examples in figure 8 cherry picked or representative of the pixel-wise uncertainty rejection sampling method ?

2. Figure 2 tries to demonstrate that despite skipping the "pixel-variance sum" statistic is able to separate out high uncertainty images from low uncertainty images. However at skipping=3 and skipping=4 the two clusters seem to be mixed quite a lot. Also it's not clear why the mean of the scores decreases for skipping=5 and skipping=6 when it was increasing monotonically from skipping=1 to 4.

---

> ### Author Response · Authors · 2023-11-16
> **Response to Reviewer NYDr**
>
> Thank you for the positive feedback and useful suggestions! We are glad you thought "this paper is reasonably novel and presents a methodology that practitioners may find useful". We address your concerns point by point below.
>
> Q1:  Questions about qualitative results for Section 4.2.
>
> The flawed samples are identified by humans and the bounding boxes are manually annotated. This is reasonable and feasible due to a typical pattern of chat logs in the Discord server of Stable Diffusion: users provide a prompt, and the channel bot generates multiple images based on the prompt. Subsequently, individuals manually select one image for further processing [1]. We have included additional prompts and corresponding images containing artifacts annotated by humans in Appendix A.7. By observing the resampling results, we find that the success rate of resampling flawed samples into desirable samples is approximately at least 60%. We will involve more human experts in the evaluation process in the final version.
>
> Q2:  Consistency between BayesDiff and BayesDiff-Skip.
>
> For samples with uncertainty around the mean, it is indeed challenging to differentiate them steadily. However, as mentioned in the article, the top and bottom 10% of samples based on uncertainty can be correctly classified. We compare the top and bottom 10% images based on the highest and lowest uncertainty in the new version of Figure 2, which fulfills the requirement of accurately excluding samples with the highest uncertainty in practical applications. Furthermore, we have directly visualized the sorting of samples with different skipping sizes in Appendix A.6. These visualizations demonstrate that regardless of the skipping size, there is a consistent trend from cluttered to clean images. This finding confirms that skip algorithms can be employed in practical applications.
>
> The uncertainty dynamics in diffusion models are complex, as the relationship between Cov$(x_t, \epsilon_t)$ and Var$(\epsilon_t)$ with respect to $t$ is not necessarily a simple monotonic relationship. Therefore, we cannot consider the length of the skipping interval as a simple monotonic relationship with the sum of pixel-wise variance. The exact relationship between them is not easily determined through theoretical analysis. However, this does not hinder our application of skipping algorithms.
>
> Given these improvements and changes to your initial concerns, we hope for your reconsideration in raising your score.
>
> [1] Wu X, Sun K, Zhu F, et al. Better aligning text-to-image models with human preference. ICCV 2023.

---

### Official Review · Reviewer_B5Gy · 2023-10-31

**Soundness:** 2 fair
**Presentation:** 3 good
**Contribution:** 2 fair
**Rating:** 6
**Confidence:** 3

**Summary:**

Authors propose to obtain a Laplace approximation to the last layer of a diffusion model to filter out low-fidelity images.

**Strengths:**

**Originality.** The idea of obtaining Laplace approximations to the weights of neural networks is not new, neither is the idea of filtering out low-fidelity images.

**Quality and clarity.** The paper is easy to follow.

**Significance.** I do not find the proposed approach a theoretically sound approach, hence not significant.

**Weaknesses:**

* Why not use the model likelihood to rule out low-fidelity images? The likelihood of diffusion models is tractable as done in [Song et al.](https://openreview.net/pdf/ef0eadbe07115b0853e964f17aa09d811cd490f1.pdf)

* Despite authors' justification, I am not convinced that the posterior distribution over the weights of the last layer can be accurately approximated with a Gaussian distribution. This statement is as accurate as the statement that a Gaussian prior is a good prior for the weights of a neural network. Is that true? I suggest plotting per-weight histograms of the last layer of a trained diffusion model to see if they are Gaussian.

* The whole notion of removing low-fidelity images and promoting generative models to create "good looking" images has been recently highly criticized due to this process biasing generative models. See [this paper](https://arxiv.org/pdf/2106.10270.pdf) and [this paper](https://arxiv.org/abs/2306.06130) and similar papers (in reference and citations).

**Questions:**

See weaknesses.

---

> ### Author Response · Authors · 2023-11-16
> **Response to Reviewer B5Gy (Part 1/2)**
>
> We sincerely thank the reviewer for the time to read our paper. We are glad you thought our paper was easy to follow.
>
> To address your concerns, we would like to clarify our contributions first. Obtaining Bayesian uncertainty through Laplace approximation is not the goal of this work. Instead, we target estimating the generations' uncertainty arising from Bayesian model uncertainty, which has not been investigated before. We overcome the challenges caused by the complexity of the iterative sampling process and the gap between the uncertainty of model output (estimated score function) and the uncertainty of sample $x_0$ in the diffusion models. Our empirical achievements involve not only the filtering of low-fidelity generation but also the data augmentation and error rectification ability in text-to-image tasks. Next, we address your feedback point by point below.
>
> W1：Why not use the model likelihood to rule out low-fidelity images?
>
> Thank you for providing this insight, but as far as we know, a model can have poor log-likelihood and produce great samples, or have great log-likelihood and produce poor samples. **Log-likelihood and visual appearance of samples are therefore largely independent.** You can refer to this article [[1](https://arxiv.org/abs/1511.01844)] for details.
>
> Based on [the official code of the paper you provided](https://github.com/yang-song/score_sde_pytorch), we computed the per-sample negative log-likelihood (NLL) w.r.t the ODE reverse process, i.e., $- log (p_\theta^{ODE}(x_0))$ for two datasets. Detailed experimental results can be found in Appendix A.5. **Experimental results demonstrate that model likelihood is indeed unsuitable for filtering out low-quality images. In comparison, the sample-wise metric based on uncertainty enables us to effectively eliminate low-quality images with cluttered backgrounds.** Due to time constraints, we are currently in the process of measuring the quantitative metric for comparison, which will be added to the article as soon as we complete it.
>
> W2：The confusion about the posterior distribution of the parameters.
>
> If we understand correctly, when you mention "per-weight histograms of the last layer of a trained diffusion model", you are referring to plotting the posterior distribution of the parameters of the last layer of the pre-trained model after introducing Bayesian priors to these parameters.
>
> To address this concern, we would like to clarify that in our setting, where we introduce Gaussian prior to the NN parameters, **the posterior distribution of the last-layer parameters is exactly a Gaussian distribution rather than being approximated as one.** Note that:
>
> - The Gaussian prior is common in practice. E.g., the widely used weight decay regularization in optimization just corresponds to the MAP estimation under a centered Gaussian prior.
> - The training objective, in the form of the L2 norm, actually corresponds to modeling the likelihood probability $p(y|f_\theta(x))$ as a Gaussian distribution.
> - For the parameters of the last layer (denoted as $\theta_L$) of the network, $p(\mathcal{D}|\theta_L)= p(y|f_{\theta_L}(x))$ is still Gaussian distribution regarding $\theta_L$.
>
> Therefore, $p(\theta_L| \mathcal{D})\propto p(\theta_L)p(\mathcal{D}|\theta_L)$ is exactly a Gaussian distribution.
>
> Additionally,  to address your concerns, we plot posterior distributions of the first four parameters of the last linear layer of U-ViT, which contains 18864 parameters in total, in Figure 11 of Appendix A.4.
>
> [1] Theis, L., A. van den Oord, and M. Bethge. A note on the evaluation of generative models. International Conference on Learning Representations 2016.
>
> [2] Song Y, Durkan C, Murray I, et al. Maximum Likelihood Training of Score-Based Diffusion Models. Advances in Neural Information Processing Systems 2021.

---

> ### Author Response · Authors · 2023-11-16
> **Response to Reviewer B5Gy (Part 2/2)**
>
> W3：The whole notion of removing low-fidelity images and promoting generative models to create "good looking" images has been recently highly criticized due to this process of biasing generative models.
>
> Firstly, thank you for providing the paper for discussion, which highlights the potential undesired effects of incorporating AI-created data into the training dataset of real data, such as inducing bias in future versions of generative models. It has inspired insights into the future development of generative models. However, **it is important to clarify that our paper does not aim to promote filtered data as new valuable data for training or finetuning generative models, thus we will not induce bias into generative models.**
>
> Instead, our objective is to derive a sample-wise metric based on uncertainty which can be utilized to detect low-fidelity data generated by a pre-trained model. This is important and meaningful for the application of large-scale generative models, which often suffer from low-quality generations [3, 4]. By doing so, we can enhance the quality of feedback provided to users, thereby demonstrating practical significance. We hope that this restatement of our motivation addresses your concerns.
>
> We hope these reclarifications and explanations of our method in response to your initial concerns can convince you to increase your score.
>
> [3] Ma Y, Yang H, Wang W, et al. Unified multi-modal latent diffusion for joint subject and text conditional image generation. arXiv:2303.09319, 2023.
>
> [4] Tang Z, Gu S, Bao J, et al. Improved vector quantized diffusion models. CVPR 2022.

---

> > ### Author Response · Authors · 2023-11-21
> > **Sincerely looking forward to the further discussions**
> >
> > Dear reviewer,
> >
> > We are wondering if our reclarifications and explanations of our method have resolved your concerns. If our response has addressed your concerns, we hope for your reconsideration in raising your score.
> >
> > If you have any additional questions or suggestions, we would be happy to have further discussions.
> >
> > Best regards,
> >
> > The Authors

---

> > > ### Comment · Reviewer_B5Gy · 2023-11-21
> > >
> > > Thank you for your thorough responses. My concerns are mostly addressed. I will consider raising my score.

---

> ### Author Response · Authors · 2023-11-22
> **Acknowledgement**
>
> Thank you for your feedback！

---

### Author Response · Authors · 2023-11-16
**General Response**

We appreciate that reviewers acknowledge that our method is “reasonably novel and presents a methodology that practitioners may find useful” (Reviewer NYDr), “there may not have been any prior work in this direction” (Reviewer CcvE), and our method “sound and most importantly, very practical” (Reviewer AU5E). We thank all reviewers for their elaborate and constructive feedback, which has led to a substantially stronger paper (with the revision now posted).  We made the following changes to our paper:
- Direct visualization of sorting based on uncertainty estimated by BayesDiff is added to Appendix A.6.
- Additional experimental results regarding Section 4.2 are added to Appendix A.7.
- Error bar is added to the FID results in Table 1.
- We adjust the color palettes of Figure 2 and Figure 5 to make them clearer.

We hope that with these changes, we address all major concerns.

---

### Meta-Review · Area_Chair_UD6u · 2024-01-07

**Metareview:**

Everybody votes in favor of acceptance after a convincing author response phase.

**Justification For Why Not Higher Score:**

mostly borderline votes

**Justification For Why Not Lower Score:**

all in favor of acceptance

---

### Decision · Program_Chairs · 2024-01-16

Accept (poster)